# The Rural Livability Evaluation and Its Governance Path Based on the Left-Behind Perspective: Evidence from the Oasis Area of the Hexi Corridor in China

Libang Ma [1,2,3,*] , Yuqing Zhang [1], Zhihao Shi [1] and Haojian Dou [1]

[1] College of Geography and Environmental Science, Northwest Normal University, Lanzhou 730070, China; zhangyq219@126.com (Y.Z.); geographyszh@163.com (Z.S.); 18394136269@163.com (H.D.)
[2] Key Laboratory of Resource Environment and Sustainable Development of Oasis, Lanzhou 730070, China
[3] Northwest Institute of Urban-Rural Development and Collaborative Governance, Lanzhou 730070, China
[*] Correspondence: malb0613@nwnu.edu.cn; Tel.: +86-931-797-1754

**Abstract:** The evaluation of rural livability for different groups of left-behind people and proposing classified governance paths are of great practical significance to solve the problem of sustainable development of left-behind villages. Taking Jinchang, China as an example, this paper aims to construct a rural livability evaluation index system based on identifying the types of left-behind villages, which combines the "individuality + commonality" of different left-behind subjects, analyzes the livability level of left-behind villages and proposes a classified governance path to help solve the problem of sustainable development of left-behind villages. The results show the following: (1) The types of left-behind villages are mainly left-behind children and left-behind elderly types, accounting for 68.75% of the total number of left-behind villages. (2) There are large differences in the livability of individual characteristics of the villages. The average livability for children is the largest, reaching 0.6608. The average livability for women is the smallest, being only 0.1418. The livability values for the elderly and children are mainly in the medium-value areas, while the livability for women is mainly in the low-value areas. (3) The overall livability level of the villages is low, mainly falling in the low-value areas. The evaluation units with values higher than the average accounted for 40.625% of the total. The level of meeting the demands of the left-behind population in villages is low. The overall levels of economic development, public services, infrastructure, and configuration need to be optimized and improved, and the living and production conditions need to be further improved. (4) According to "left-behind + livable", we classified the villages into five types: optimizing and upgrading villages, improving short-board villages, balanced developing villages, upgrading potential villages, and comprehensive upgrading villages. In the future, it is necessary to carry out classified governance from various aspects, such as improving governance, making up for shortcomings, coordinating and balancing, and comprehensively improving quality to achieve the ultimate goal of sustainable rural development.

**Keywords:** left-behind perspective; "three-stay" population; rural livability; governance path; Jinchang city; China

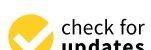



## 1. Introduction

### 1.1. Research Background

Globalization, informatization, and urbanization are driving the continuous transformation and reshaping of urban-rural relations. Global urban-rural relations have experienced a process from urban-rural division and opposition to urban-rural coordination, integration, and equivalence [1]. China's urban and rural areas are characterized by interdependence of economic development, complementarity of resources, and the symbiosis of ecology [2]. However, the long-term siphon effect of cities on rural areas has resulted in many rural problems, such as non-agricultural and hollowing out of production factors,

the aging and impoverishment of rural subjects, and pollution of the rural environment [3]. Under the background of the rural revitalization strategy, the goal of sustainable rural development is to have a suitable and beautiful natural environment, convenient transportation, a good employment environment, rich culture, and excellent medical and educational resources. The countryside is a regional complex with natural, social, and economic characteristics. At the current stage, the development of the countryside needs not only the economic level to measure but also production, life, ecology, culture, and other aspects, as well as different subject perspectives to be evaluated. Therefore, the key to achieving sustainable rural development is to scientifically construct an evaluation index system for livable villages from a microscopic perspective and to put forward an accurate and reasonable classified governance path according to the type of village development.

In recent years, there have been many examples of research on the urban and rural living environments and their livability. In 1954, the Greek architectural planner Dossadias first proposed the theory of "human agglomeration". He mentioned "human beings are no longer satisfied with their living environment" in his book *Introduction to the Science of Human Habitat* in 1968. This opened up research on human settlements. In 1961, The World Health Organization (WHO) put forward the concept of the safety, health, convenience, and comfort as the core of human settlements [4]. Overall, the research on the livability of urban areas is well-developed with relatively complete evaluation system and systematic research methods [3,5–11], but the research on the livability of rural areas is not. Early research on rural livability mostly focused on the natural environment and ecological security, and the research methods were mostly subjective descriptions [12–14]. In the middle and late 20th century, the focus of rural livability research gradually shifted from nature to rural economic development, social sciences, and humanities because of more and more frequent interactions between behaviors, sociology, economics, and other disciplines and geography. The scientific results from multiple perspectives, fields, and disciplines have begun to be widely applied in the specific research on rural livability. Scholars generally believe that factors such as population density, immigration, and rural public policies are the main factors affecting rural livability [15–17]. The research on rural livability in China began with the "science of human settlements" proposed by academician Liangyong Wu at the end of the 20th century [18]. After that, with the progress of the measurement revolution, behavioral revolution, and computer science and technology, most of the research on rural livability is a systematic and comprehensive evaluation that is mainly based on a combination of qualitative and quantitative methods. The specific research objects are mostly concentrated on the intensive utilization of rural land and the rural landscape environment [19–29].

### 1.2. The Current Situation of the Left-Behind Population in China

For a long time, due to rapid urbanization, most of China's rural areas have faced serious problems such as hollowing out, marginalization, aging and weakening, cultural inheritance faults, destruction of historical features, and defacement of natural features [28,30–33]. Rural decline urgently needs to determine an effective method for rural revitalization and development from the perspective of urban-rural dynamics [34], emphasizing the integration of urban-rural development space and optimizing the distribution pattern of rural areas [35,36] so as to build a bridge for a virtuous circle between urban and rural areas [37] and achieve sustainable development in the context of coordinated urban and rural relations [38,39]. In the report of the 19th CPC National Congress in 2017, the CPC clearly put forward a rural revitalization strategy and the general requirements of "prosperous industry, livable ecology, civilized rural customs, effective governance, and affluent life" to implement a rural revitalization strategy, build beautiful and livable villages, and give full play to the main role of farmers. The government should focus on those farmers who stay in the countryside and do not enter the city, especially the rural left-behind population (referring to the family core labor force going out to work in the process of urbanization and being left in the countryside due to subjective and objective

conditions, including the basic needs of the elderly, women, and children [40]). In 2020, the scale of China's floating population reached 376 million, accounting for 26.63% of the total population. The rural left-behind population reached 87 million, accounting for 17.06% of the total rural population. The problem of the left-behind population in China's rural areas has become increasingly prominent, and it has become an overall problem that must be addressed in the period of social transformation (a period of profound changes in socioeconomic structures, cultural patterns, values, etc.). However, the rural left-behind population has become the "forgotten group" in urban and rural development. Society pays less attention to the rural left-behind population. As the main population living in the countryside for many years, the livability of the rural left-behind population cannot be ignored. Therefore, improving the living environment in rural areas and building beautiful and livable villages for the elderly, women, and children has become an important task for China to implement the rural revitalization strategy. People are the main body of rural revitalization. Although the existing research has paid attention to the main element of people, most of them regard all villagers as the main body of rural livability evaluation, ignoring the different demands of different groups. The standards of livability are also different. The left-behind elderly, women, and children have different demands for village livability because of their differences in age, gender, education level, and attitude toward society. As the recipients of changes in urban-rural relations, the left-behind population is relatively weak in comprehensive ability. It is difficult to substantively participate in the process of rural development and rural governance. This leads to problems such as the absence of a main body of governance and a lack of governance in rural governance, and they face challenges from both inside and outside families. China is at a critical stage of effective connection between poverty alleviation and rural revitalization. During the period of social transformation, the problems of "agriculture, rural areas, and farmers" are prominent and urgently need scientific support. Therefore, it is of great practical significance to evaluate the livability of villages for different groups of left-behind people and propose a classified governance path for solving the problem of sustainable development of left-behind villages.

In the process of agricultural modernization, the rural labor force in Jinchang has moved out of the countryside in order to seek better development opportunities. The number of labor exporters is large, and the phenomenon of rural left-behind people is serious. The left-behind population accounts for 3.72% of the total rural population. The data from the Seventh Census in 2020 shows that compared with the sixth census in 2010, the urban population of Jinchang city increased by 50,840, the rural population decreased by 76,864, and the proportion of the urban population increased by 15.30%, which was 1.09% higher than that of the national urban population increase. In terms of population flow, it represents the situation in most rural areas in China. Meanwhile, Jinchang is in an ecologically fragile area where deserts and oases meet, and the population is mainly distributed around a few oases. The higher food production and greater number of plants in the oases provide good natural environmental conditions for agricultural production and residential life, and the research and exploration of sustainable development of the oases can provide more appropriate guidance for the construction of livable villages. Therefore, this paper aims to take Jinchang, Hexi Corridor, China as an example, identifying the left-behind types of villages by using rural survey data and evaluating rural livability by establishing an evaluation index system with "individuality + commonality". Combining the characteristics of left-behind and livability, this paper divides different types of villages and proposes corresponding strategies for optimizing governance. These are of great significance to promote rural revitalization and urban-rural integrated development.

## 2. Theoretical Analysis

Since the implementation of the household contract responsibility system in China (a form of agricultural production responsibility in which farmers contract land from villages or groups on a household basis), rural productive forces have been liberated.

The control function of the household registration system has begun to weaken, making it possible for the rural labor force to move between urban and rural areas [3]. A large number of rural populations moved into cities and towns, resulting in the increasingly serious rural hollowing phenomenon, which is characterized by extensive use of land resources, overpopulation, and insufficient economic development momentum [29,33]. The population shows a trend of aging and weakening. Young laborers move to cities, and the others left behind in rural areas are called left-behind populations. The elderly, women, and children are the main groups of the left-behind population [41]. Among them, the left-behind elderly refers to elderly people ages 60 and above. They stay in their place of residence for a certain period of time, and their children work far away from their hometown [42–44]. Left-behind women refer to women under the age of 60 whose husbands go out to work and live in the countryside by themselves [45,46]. Left-behind children refer to children or adolescents under the age of 18 who have one or both parents working far away from home. Those children live in rural areas and attend school by themselves [47–51]. The family is the basic unit for the socialization of children, the function of marriage, and caring for the elderly. However, the outflow of the main labor force in rural families has resulted in changes in the family structure. Left-behind women have become the key labor force of left-behind families, educating children and caring for the elderly. Left-behind children become the spiritual core of left-behind families and the care objects of the left-behind elderly and left-behind women.

Rural livability is a concentrated expression of the development opportunities and external conditions that villages can bring to villagers [19]. At the end of 2018, China implemented an evaluation for the construction of beautiful villages (GB/T 37072-2018) to conduct a comprehensive evaluation of beautiful countryside construction through four aspects: village construction, ecological environment, economic development, and public services. The left-behind population suffers from economic, livelihood, and mental pressures in the process of rural development and is a vulnerable group in need of care. Their development demands cannot be ignored. By accurately grasping the development needs of the left-behind population and constructing a relevant indicator system for the level of rural livability, this paper reveals the current shortcomings in rural development, identifying the important and difficult points of work. This can effectively guide rural livability construction. Maslow's hierarchy of needs states that five categories of human needs dictate an individual's behavior. Those needs are physiological needs, safety needs, social needs, esteem needs, and self-actualization needs. A person can only move on to addressing the higher-level needs when their basic needs are adequately fulfilled. The satisfaction status of each level of demand presents a pyramid state. The lower the level of demand, the better the satisfaction status. Maximizing the satisfaction of the needs of the left-behind population can enhance the "sense of satisfaction and gain" of the left-behind population and improve livability in the rural areas. A comfortable and healthy living environment, convenient and accessible public services, and urban-rural integrated infrastructure are manifestations of a villager's demand for physical and safety needs and a social life. However, the left-behind groups have different development needs and living environment needs due to their different ages and social roles. Compared with ordinary rural residents (referring to the rural non-left-behind population), they have a lower ability to resist risks, develop, and construct and need more attention paid to their development needs. The current family elderly care model is gradually changing to the new model of mutual assistance in rural society. The physical and mental health of left-behind elderly people is often fragile in the absence of children [40]. They urgently need pension security and medical care. Left-behind women, as an important labor force in families, are an easily forgotten group. They face the pressure of supporting the elderly, raising children, and working for a living. They need a good employment environment and an equal social environment with non left-behind villagers, other male villagers, and urban residents to exert their influence and value. Left-behind children lack a family education accompanied by their parents during their growth. Compared with non-left-behind children, they often have more growth risks,

psychological burdens, health problems, and behavioral problems [47]. Support from teachers and schools is important for the development of left-behind children. This can reduce anxiety in the lives of left-behind children by compensating for the lack of emotional support from their parents (Figure 1) [50].

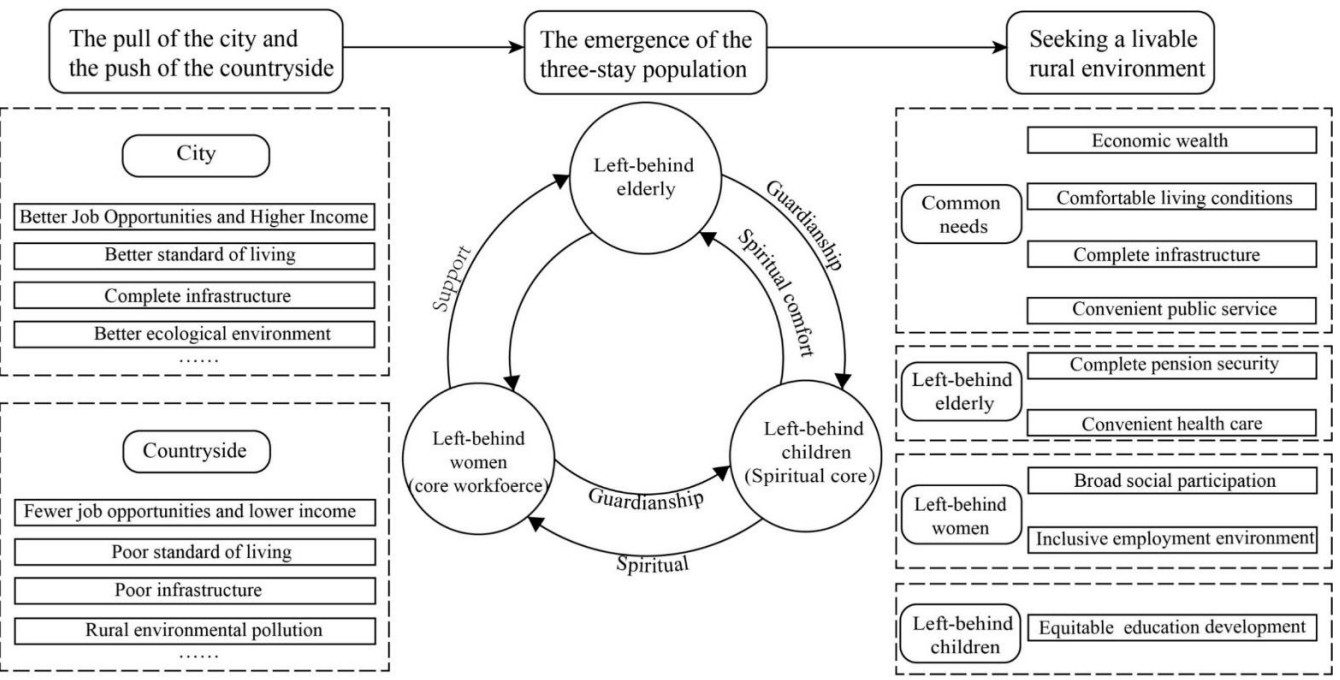

**Figure 1.** Theoretical analysis of left-behind population and rural livability.

## 3. Overview of the Study Area

### 3.1. Overview of Physical Geography

Jinchang city is located at 101°04′35″~102°43′40″ east longitude and 37°47′10″~39°00′40″ north latitude. It is located in the eastern section of the Hexi Corridor and north of the Qilian Mountains in Gansu Province, China, with a total area of 9593 km². The terrain of Jinchang city is high in the south and low in the north, and mountains, plains, and rivers are interlaced with the Gobi Oasis. The climate of Jinchang city is a continental temperate arid climate, with sufficient sunlight, an arid climate, and northwesterly winds throughout the year. There are large temperature differences throughout the four seasons. The temperature of Jinchang is higher in the north and lower in the south. Precipitation increases from the northeast to southwest as the terrain rises, with less rainfall in the gullies and more rainfall in mountainous areas. Jinchang city is located in the Shiyang River Basin, the three major inland rivers in the Hexi Corridor, being located in an inland arid area. Surface water comes from precipitation in the mountainous areas and melted snow and ice in the mountains. The groundwater is widely distributed in the southern grasslands, western grasslands, and northern grasslands, but the amount of water is very small. The replenishment of groundwater mainly depends on the underflow of mountain valleys, piedmont river channels, and rainfall infiltration. Jinchang city is one of 110 key water-scarce cities and 13 resource-based water-scarce cities in China. It is also an area with a relatively fragile natural ecological environment in western China (Figure 2).

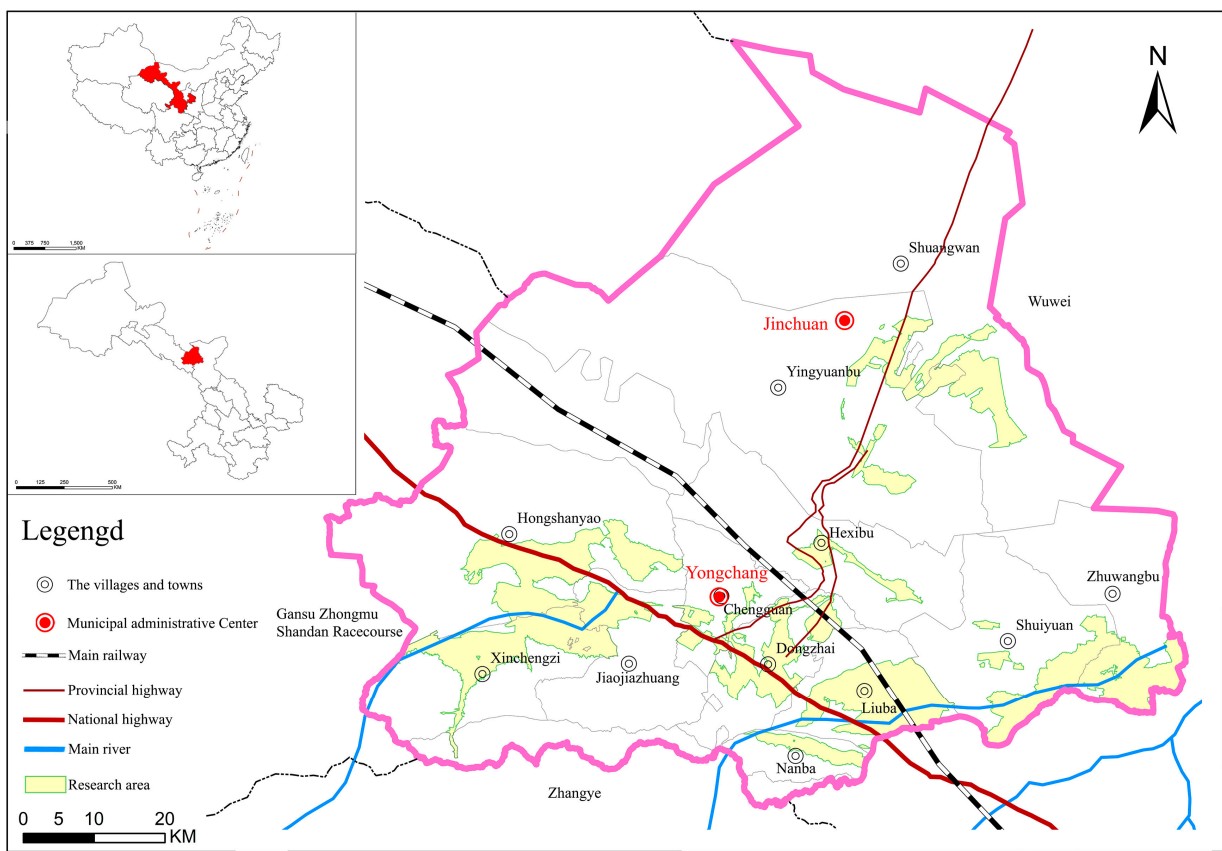

**Figure 2.** Study area map.

*3.2. Socioeconomic Overview*

Jinchang city has jurisdiction over 2 counties (districts), 12 townships, and 139 administrative villages. In 2020, the annual gross regional product was CNY 35.86 billion. The resident population of Jinchang is 438,000, of which the rural population is 99,000. The urbanization rate reached 77.40%, much higher than the average level of Gansu Province. The per capita disposable incomes of urban and rural residents were CNY 43,390 and CNY 16,976, respectively. The Engel coefficients of them were 31% and 29.6%, respectively. At the end of the year, there were 0 rural poor people in Jinchang, and the incidence of rural poverty was 0%. The number of surplus urban and rural labor transfers in Jinchang was 80,500. In 2020, among the 139 administrative villages in Jinchang city, 96 administrative villages had left-behind populations, accounting for 70.07% of the total administrative villages. The "three-stay" population was 3685, accounting for 3.72% of the total rural population. Among them, the elderly, women, and children totaled 2371, 1151, and 163, accounting for 64.34%, 31.24%, and 4.42% of the total left-behind population, respectively. As the number of migrant workers continued to increase, there were more and more problems in Jinchang city, such as pensions and medical care for the left-behind elderly in the Jinchang villages, the economic and family rights and quality of life of left-behind women, and the growth and education of left-behind children. This resulted in enormous pressure on rural social development.

## 4. Research Methods and Data Sources

### 4.1. Data Sources

The data in this article mainly came from three sources: (1) basic drawings, including a topographic map of Jinchang city and the boundary of the vector administrative village of Jinchang city (1:250,000) from the Bureau of Surveying and Mapping of Gansu Province. (2) Socioeconomic data, namely the data on the infrastructure, public service facilities,

industrial development, and living environments of various administrative villages in Jinchang city in 2020, were obtained through field investigations. Based on the Participatory Rural Appraisal (PRA), we conducted a 10-day village survey in Jinchang city in May 2021 for 11 factors, such as the population, arable land, industries, social security, facilities, and residents' lives in 139 administrative villages. (3) POI data, or the data of facilities such as hospitals, kindergartens, primary schools, junior high schools, and high schools, were from the Baidu map POIs. The distances between villages and POIs were obtained through nearest neighbor analysis in ArcGIS.

*4.2. Research Methods*

4.2.1. Participatory Rural Appraisal (PRA)

PRA is a social science survey method that analyzes the behavior of farmers and peasants under conditions of intervention in rural communities to understand rural life, rural socioeconomic activities, and rural and community development issues and opportunities. It absorbs the research methods of mathematics, economics, sociology, anthropology, surveying and mapping, and other disciplines to form a relatively standard method system. In light of the social problems of the left-behind population in Jinchang city, the research group conducted a 10-day village survey in Jinchang city in May 2021 based on the PRA method. On the basis of a comprehensive grasp of the village's natural foundation, development strengths and weaknesses, and social and public opinion, semi-structured interviews were conducted with members of the village cadres, mainly to understand the history of village development changes, the current situation and structure of the population, population status and structure, land use status and structure, and social and economic development status, as well as the basic situation of farm households (the current situation of farm households' living conditions, public health, and service facilities). The survey process strictly followed the ethical principles of informed recognition, equality, respect, and non-impairment in research, and it was completed in a communicative and interactive manner. The survey finally obtained valid data in 11 aspects, including the population status, industrial development, residents' livelihood, social security, infrastructure construction, and village collective economy of 139 administrative villages by 2020. Among them, 17 related variables were extracted from 5 aspects of industrial development, social security, infrastructure, the village collective economy, and grassroots administration for the establishment of indicators. (Table 1).

**Table 1.** Descriptive statistics of main variables.

| Variable Types | Variable | Mean Value | Standard Deviation |
|---|---|---|---|
| Social security | Number of mutual-aid elderly care service facilities owned per capita | 0.22 | 0.38 |
| | Number of health rooms per capita | 0.38 | 0.56 |
| Industrial development | Number of agricultural processing enterprises in the village | 0.54 | 1.04 |
| | Number of agricultural enterprises in the village | 0.50 | 0.84 |
| | Number of farm households with business licenses and carrying out leisure agriculture and rural tourism | 0.56 | 1.74 |
| | Number of agricultural households selling agricultural products online | 0.15 | 0.47 |
| Grassroots administration | Percentage of female cadres among village cadres | 0.25 | 0.13 |
| Village collective economy | Per capita disposable income of rural residents (1. below CNY 3000; 2. CNY 3000–5000; 3. CNY 5000–10,000; 4. CNY 10,000–15,000; 5. CNY 15,000–20,000; 6. CNY 20,000 or more) | 4.58 | 0.73 |
| | Per capita income (dividends) received by residents of this village from the village collective | 567.13 | 3455.77 |

**Table 1.** *Cont.*

| Variable Types | Variable | Mean Value | Standard Deviation |
|---|---|---|---|
| Infrastructuree | Area of homestead per number of households | 817.10 | 1197.90 |
| | Number of village groups with road access per number of village groups in the village | 0.91 | 0.21 |
| | Number of villager groups who have completed toilet renovation per number of villager groups in this village | 0.64 | 0.86 |
| | Number of villager groups connected to natural gas per number of villager groups in the village | 0.05 | 0.22 |
| | Number of villager groups connected to broadband Internet per number of villager groups in the village | 0.92 | 0.23 |
| | Number of farmers' amateur cultural organizations per capita | 0.39 | 0.75 |
| | Number of library (hall) and cultural stations per capita | 0.31 | 0.37 |
| | The number of sports and fitness venues per capita | 0.38 | 0.67 |

### 4.2.2. Identification Method of Left-Behind Village Types

This paper identified the left-behind types in different evaluation units through the pyramid model to further show the status of the left-behind people in each evaluation unit and implement rural governance countermeasures more accurately [29]. First, a rural left-behind pyramid model was constructed based on the proportion of the left-behind elderly, left-behind women, and left-behind children in the total left-behind population of each evaluation unit. Second, according to the classification standard in the pyramid model, the 96 villages in Jinchang City were divided into four types: the left-behind elderly type, left-behind women type, left-behind children type, and comprehensive type. When the left-behind elderly accounted for 50% or more of the total left-behind population, the evaluation unit was dubbed the left-behind elderly type (tower top, tower bottom wide). When the left-behind women accounted for 50% or more of the total left-behind population, the evaluation unit was deemed the left-behind women type (width of tower body, width of tower spire and tower bottom). When left-behind children accounted for 50% or more of the total left-behind population, the evaluation unit was given the left-behind children type (the spire was wide, and the bottom of the tower was sharp). When the proportions of left-behind children, left-behind women, and left-behind elderly were equal, or two of them accounted for a relatively high proportion while the other one was relatively small, the evaluation unit was given the comprehensive type.

### 4.2.3. The Livability Evaluation for the Elderly, Women, and Children

1.  The Construction of the Evaluation Index System

Constructing a reasonable and comprehensive evaluation index system is the key and quantitative basis for the accurate evaluation of the livability of the villages for the elderly, women, and children in Jinchang. First, different left-behind groups face different problems, and their demands for the living environment are also different. In addition, different left-behind groups live in the same environment and face the same living environment. Based on this, when constructing the evaluation index system, different personality indicators should be established according to different left-behind groups, and common indicators should be established in consideration of the overall living environment in rural areas.

Following the principles of scientific, systematic, and comprehensive data selection, as well as the requirements of policy relevance, multi-dimensional comprehensiveness, research object relevance, and data accessibility of rural livability, and referring to the evaluation standards of rural livability at home and abroad, individual indicators were established from the supply of their own needs from the environment of the left-behind elderly, left-behind women, and left-behind children (Table 2). Common indicators were established from the perspective of the common demands of the rural population, such as economic affluence, living conditions, infrastructure and public services (Table 2). Young

and middle-aged rural people enter the city for economic interests. Their parents are left in the countryside because of their age, forming a group of left-behind elderly people [32]. The survey found that the rural left-behind elderly often live simple lives with a poor living environment. Many of them are still engaged in heavy field work, and a significant proportion of them suffer from geriatric diseases. The establishment of a livable environment for the rural left-behind population requires the support of the social and physical environment. The left-behind elderly have a high demand for elderly care and medical care. Therefore, the individual indicators of the left-behind elderly were established from two aspects: pension security and medical health. In addition, a large number of young and middle-aged male laborers work far away from their hometowns for a long time. Due to various reasons, they are separated from their spouses, forming a group of left-behind women. Due to the long-term absence of husbands, left-behind women often bear huge family pressure alone. Economic pressure is at the top of the list, but the income of many rural women is not stable, which will result in a lower level of life security and limited development opportunities for family members. Therefore, a stable employment environment is key to left-behind women solving their economic pressures. Therefore, the individual indicators of left-behind women were established from two aspects: employment environment and social participation. In addition, due to obstacles such as economic conditions and the household registration system, migrant workers have to leave their children in the countryside under the care of their grandparents, forming a large-scale group of left-behind children. The life of left-behind villages has brought serious and long-term impacts on the physical and mental growth of these children, especially in educational resources. Therefore, the individual indicators of left-behind children were established from educational development.

**Table 2.** Rural livability evaluation index system and index weight for left-behind population.

| Target Level | First-Level Indicators | Second-Level Indicators | Direction of Index |
|---|---|---|---|
| Left-behind elderly individual indicators | Pension security | Number of mutual − aid elderly care service facilities owned per capita ($X_1$) | + |
| | Medical health | Number of health rooms per capita ($X_2$) | + |
| | | Distance to Class I hospital ∗ ($X_3$) | − |
| | | Distance to Class II hospital ∗ ($X_4$) | − |
| | | Distance to Class III hospital ∗ ($X_5$) | − |
| Left-behind women individual indicators | Employment environment | Number of agricultural processing enterprises in the village ($X_6$) | + |
| | | Number of agricultural enterprises in the village ($X_7$) | + |
| | | Number of farm households with business licenses and carrying out leisure agriculture and rural tourism ($X_8$) | + |
| | | Number of agricultural households selling agricultural products online ($X_9$) | + |
| | Social participation | Percentage of female cadres among village cadres ($X_{10}$) | + |
| Left-behind children individual indicators | Educational development | Distance from kindergarten ($X_{11}$) | − |
| | | Distance from primary school ($X_{12}$) | − |
| | | Distance from junior high school ($X_{13}$) | − |
| | | Distance from high school ($X_{14}$) | − |
| Common indicators | Economic affluence | Per capita disposable income of rural residents ($X_{15}$) | + |
| | | Per capita income (dividends) received by residents of this village from the village collective ($X_{16}$) | + |
| | Living conditions | Area of homestead per number of households ($X_{17}$) | + |
| | Infrastructure | Number of village groups with road access per number of village groups in the village ($X_{18}$) | + |
| | | Number of villager groups who have completed toilet renovation per number of villager groups in this village ($X_{19}$) | + |
| | | Number of villager groups connected to natural gas per number of villager groups in the village ($X_{20}$) | + |
| | | Number of villager groups connected to broadband Internet per number of villager groups in the village ($X_{21}$) | + |

**Table 2.** *Cont.*

| Target Level | First-Level Indicators | Second-Level Indicators | Direction of Index |
|---|---|---|---|
| | Public services | Number of farmers' amateur cultural organizations per capita ($X_{22}$) | + |
| | | Number of library (hall) and cultural stations per capita ($X_{23}$) | + |
| | | The number of sports and fitness venues per capita ($X_{24}$) | + |

* Remarks: Class I, II, and III hospitals are classified according to the "Hospital Classification Management Measures" issued by the Ministry of Health.

2. Data Standardization

When evaluating livability for the elderly, women, and children of Jinchang's villages, the initial data were standardized in order to exclude the influence of different index dimensions and numerical values on the results. The larger the value of the indicator, the higher the level of livability, and the positive index calculation formula in Equation (1) was used for normalization; The smaller the value of the indicator, the lower the level of livability, and the negative index calculation formula in Equation (2) was used for standardization. The calculation formula is as follows:

$$\text{positive index}: \ Z_i = (X_i - \min X_i)/(\max X_i - \min X_i) \tag{1}$$

$$\text{negative index}: \ Z_i = (\max X_i - X_i)/(\max X_i - \min X_i) \tag{2}$$

In Equations (1) and (2), $X_i$ and $Z_i$ are the original and standardized values of the $i$-th index in 2020 for each administrative village in Jinchang city, respectively, while *max* $X_i$ and *min* $X_i$ are the maximum and minimum values of the $i$-th indicator, respectively.

3. Entropy Method

In order to avoid the bias caused by human factors, this paper adopted the entropy of objective weighting method to calculate the livability evaluation results. The entropy method is an objective method to analyze the weight, which can prevent bias caused by over-reliance on subjective feelings. It is generally believed that the greater the variation in the value of an indicator, the lower the information entropy, the greater the amount of information provided by the indicator, and the greater the weight of the indicator, and vice versa for smaller indicator weights. We have

$$\delta_i = \frac{D_i}{\overline{Z}_i} \tag{3}$$

$$W_i = \frac{\delta_i}{\sum_{i=1}^{n} \delta_i} \tag{4}$$

where $\delta_i$ $D_i$, $\overline{Z}_i$, and $W_i$ are the coefficient of variation, mean square, error, mean, and weight of the $i$-th index, respectively.

4.2.4. The Evaluation of Comprehensive Livability

According to the results for identifying the left-behind types of the villages in Jinchang city based on the left-behind pyramid model, we calculated the final comprehensive livability level of each left-behind village:

$$P = \sum_{i=1}^{5} W_i \times Z_i + \sum_{i=6}^{10} W_i \times Z_i + \sum_{11}^{14} W_i \times Z_i \tag{5}$$

$$\text{ERL} = P \times W_1 + \sum_{i=15}^{24} W_i \times Z_i \tag{6}$$

where $P$ is the individual livability level of the village, the evaluation of rural livability (ERL) is the comprehensive livability level of the village, and $W_1$ is the weight of the individual livability level. The greater the ERL value, the higher the rural livability level, and vice versa.

*4.3. The Classification of Village Governance Types*

The classification of village governance types was based on the theoretical framework to analyze the demand factors of the left-behind population in the development process. According to the spatial differentiation characteristics of the elements, such as economic affluence, living conditions, infrastructure, public services, pension security, medical health, employment environment, social participation, and educational development, we selected the comprehensive evaluation of the four subsystems of "livability for elderly, livability for women, livability for children, and common livability". According to the balance of various livable levels of the village and the results of other scholars' classifications of villages [22,29], we reasonably classified the types of villages to provide a theoretical reference for the classification to promote rural development.

## 5. Result Analysis

### 5.1. Identification of Left-Behind Village Types

In 2020, there were 96 left-behind villages in Jinchang city, including 65 villages with left-behind elderly, 41 villages with left-behind women, and 61 villages with left-behind children. The identification results of the left-behind pyramid model show that the types of left-behind villages were mainly left-behind children and left-behind elderly types, accounting for 68.75% of the total number of left-behind villages. The number of left-behind elderly villages was the largest with a total of 38, accounting for 39.58% of the total number of left-behind villages and mainly being concentrated in the western part of Yongchang County. There were 28 left-behind children villages, accounting for 29.16% of all left-behind villages, mainly in the southern part of Yongchang County and the oasis and desert intersection of Jinchuan District. There were 21 left-behind women villages, accounting for 21.88% of all left-behind villages, mainly along national highway 312 and provincial highways. The number of comprehensive left-behind villages was the least with a total of 9, accounting for 9.38% of all left-behind villages, mainly in the southeast of Yongchang County. At the county level, the left-behind villages were mainly concentrated in Yongchang County, accounting for 84.21% of all the left-behind villages. The left-behind villages in Yongchang County are mainly distributed along the national highway. In addition, as the distance increased, the number of left-behind elderly villages increased. At the township scale, Xincheng town had the largest number of left-behind villages with a total of 12, and all of them were left-behind elderly villages. The number of left-behind villages in Zhuwangbao town was second with a total of 11, but the types of left-behind villages were different (Figure 3).

### 5.2. Evaluation of the Livability of the Village

According to the evaluation index system of village livability, we calculated the individual livability, common livability, and comprehensive livability level of the left-behind villages, respectively. They were divided into three levels with the help of the natural breakpoint method: low-value areas (I-level), medium-value areas (II-level) and high-value areas (III-level).

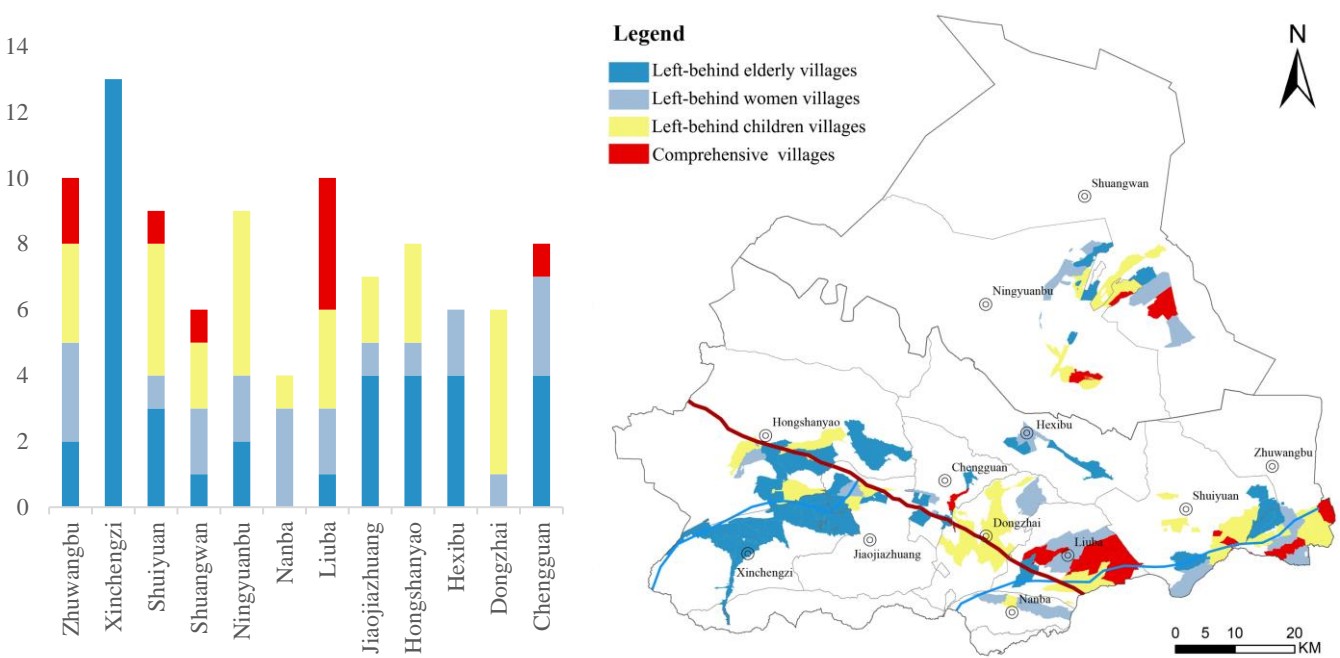

**Figure 3.** Types of left-behind villages.

5.2.1. Individual Livability

There was a big spatial difference in the level of livability for elderly in the villages. The average value of livability was 0.1874, and the above-average evaluation units accounted for 54.17% of the total number of evaluation units (Figure 4a). Among them, 44 were low-value areas, accounting for 45.83% of the total number of left-behind villages, and were scattered throughout the study area. The number of medium-value areas was the largest, reaching 47, accounting for 48.95% of the total number of left-behind villages and mainly being in the oasis-desert-steppe intersection in southern Yongchang County. The number of high-value areas was the least at only 5, accounting for 5.20% of the total number of left-behind villages and mainly being in the southeast of Yongchang County. Among them, the number of low value areas in 38 left-behind elderly villages was the highest, reaching 20, accounting for 52.63% of the total number of left-behind elderly villages. There were 17 middle-value areas, accounting for 44.74% of the total left-behind elderly villages. There was only one high value area: Xizhuangzi village in Hexibao town.

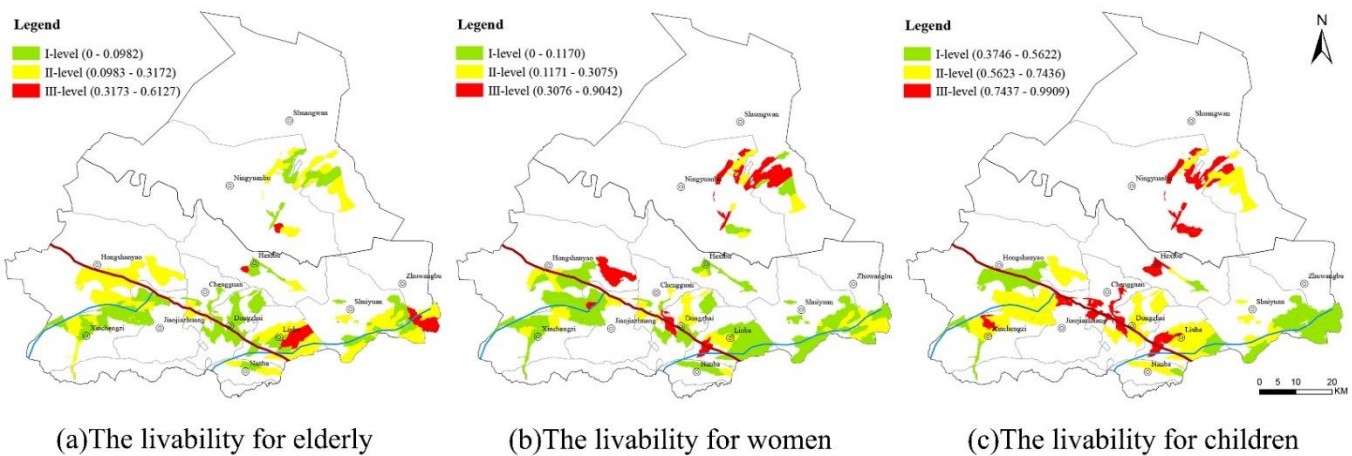

(a)The livability for elderly     (b)The livability for women     (c)The livability for children

**Figure 4.** Spatial distribution of individual livability.

The overall livability level for women was low. The average livability was 0.1418, and the above-average evaluation units accounted for 29.17% of the total number of evaluation units (Figure 4b). Among them, the number of low-value areas was the largest, reaching 58 and accounting for 60.42% of the total number of left-behind villages, which were mainly distributed in the oasis-desert-steppe intersection in southern Yongchang County. There were 28 medium-value areas, accounting for 29.17% of the total number of left-behind villages, which were scattered throughout the study area. The number of high-value areas was the least, being only 10, accounting for 10.41% of the total number of left-behind villages and mainly being distributed in Jinchuan District and the urban periphery of Yongchang County. Among them, 21 left-behind women villages had the largest number of low-value areas, reaching 14, accounting for 66.67% of the total number of left-behind women villages. There were 5 medium-value areas, accounting for 23.81% of left-behind women villages. There were only two high-value areas: Tianshengkeng village in Shuangwan town and Xipo village in Ningyuanbao town.

The livability level of children was relatively high. The average livability was 0.6608, and the minimum value was 0.3746. The above-average evaluation units accounted for 48.95% of the total number of evaluation units (Figure 4c). Among them, there were 26 low-value areas, accounting for 27.08% of the total number of left-behind villages, which were mainly distributed in the grassland belt in western Yongchang County and the desert- and oasis-interlaced belt in eastern Yongchang County. The number of medium-value areas was the largest, reaching 45, accounting for 46.88% of the total number of left-behind villages and mainly being distributed in the oasis-desert-steppe intersection in southern Yongchang County. The number of high-value areas was the least, reaching 25, accounting for 26.04% of the total number of left-behind villages and mainly being distributed in Jinchuan District and the central part of Yongchang County. Among them, 8 of the 28 children-dominant villages were low-value areas, accounting for 28.57% of the total number of children-dominant villages. The medium-value areas were the largest, reaching 14 and accounting for 50% of the total number of children-dominant villages. There were 6 high-value areas, accounting for 21.43% of the total number of children-dominant villages.

### 5.2.2. Comprehensive Livability Level

The comprehensive livability level of the village was relatively low. The average value of livability was 0.2336, and the minimum value was only 0.0341. The above-average evaluation units accounted for 40.625% of the total number of evaluation units (Figure 5). Among them, the number of low-value areas was the largest, reaching 58 and accounting for 60.42% of the total number of left-behind villages. There were 33 medium-value areas, accounting for 30.37% of the total number of left-behind villages. Both the low-value areas and medium-value areas were scattered throughout the study area. The number of high-value areas was the least, being only 5, accounting for 5.21% of the total number of left-behind villages. They were scattered in Yongchang County.

### 5.3. Classification of Governance Types in Villages

According to the individual development needs of the three types of left-behind populations and the common level of rural livability, we made a grade-by-level classification judgment (Figure 6). Based on the combination of characteristics, the types of left-behind villages were identified as optimizing and upgrading villages, improving short-board villages, balanced developing villages, upgrading potential villages, and comprehensive upgrading villages (Figure 7).

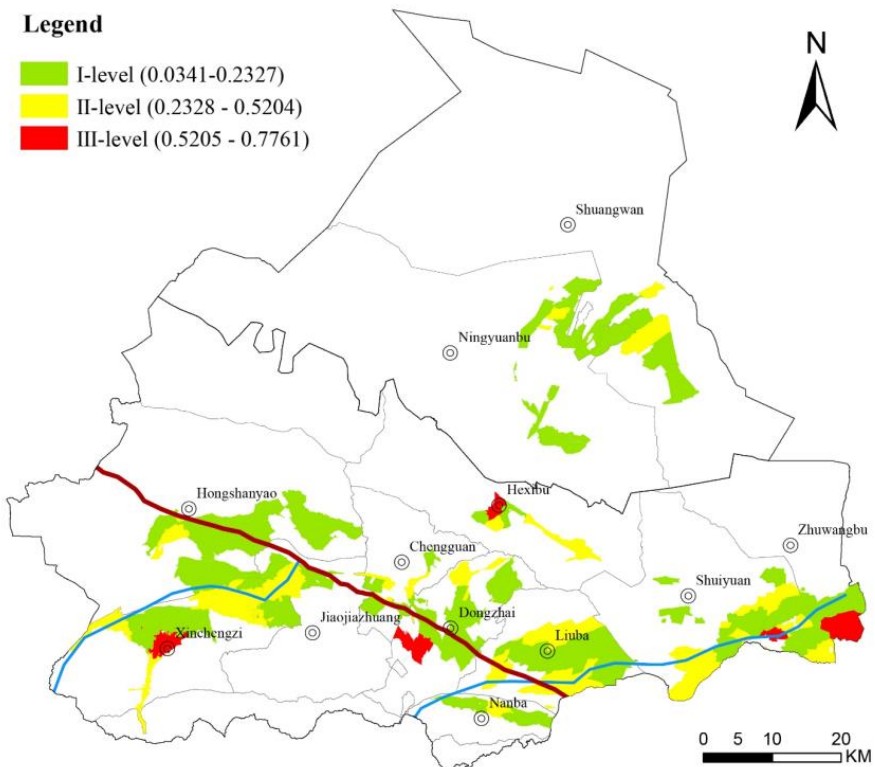

**Figure 5.** Spatial distribution of comprehensive livability.

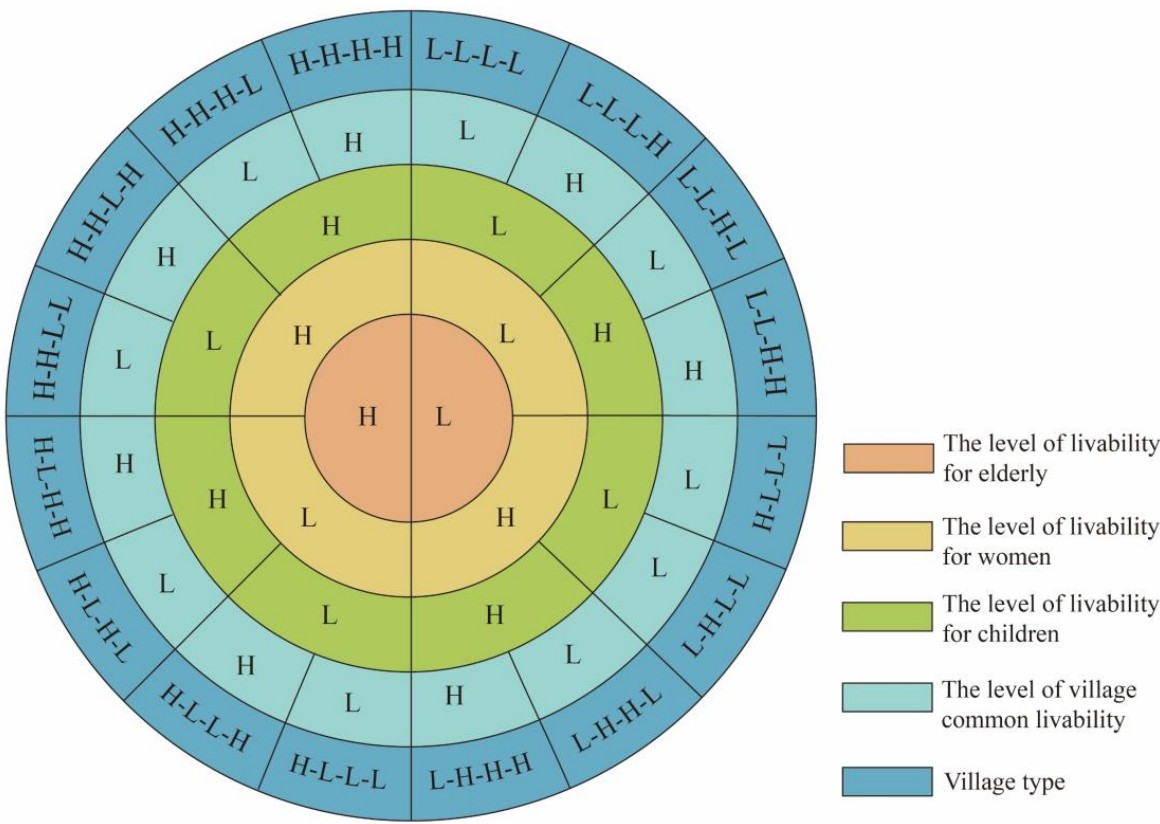

**Figure 6.** Technical route for type identification of left-behind villages.

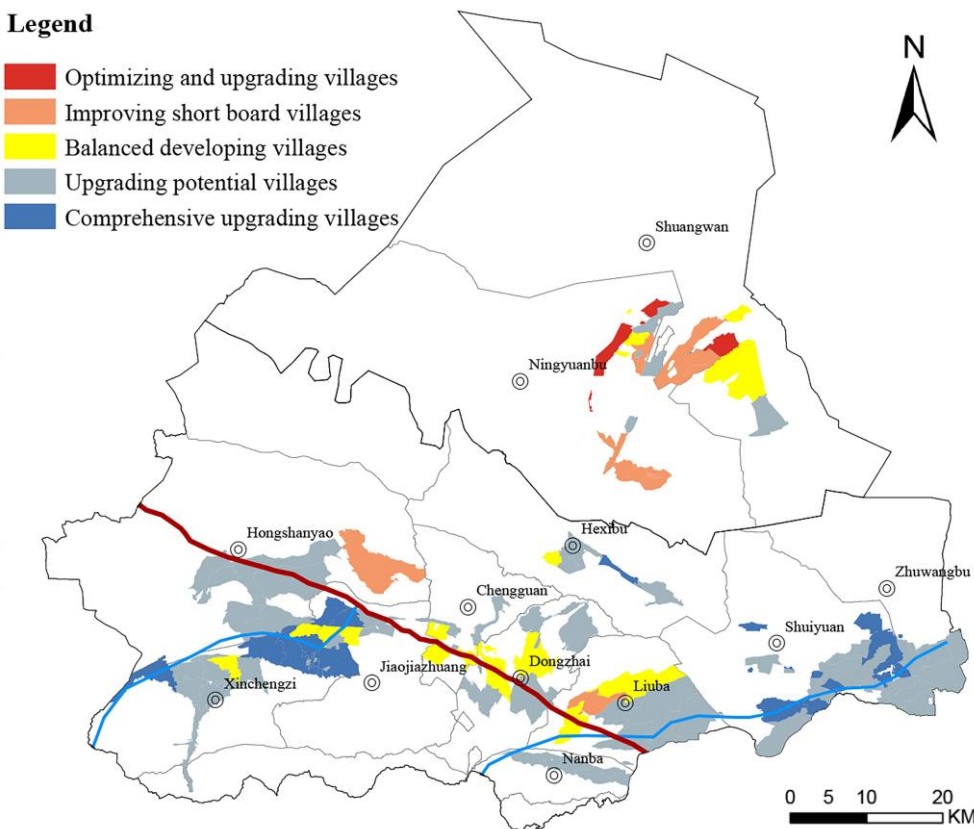

**Figure 7.** Spatial distribution of left-behind village types.

1. Optimizing and upgrading villages: This type is mainly H-H-H-H type villages. The livability for the elderly, women, and children as well as common livability are all at a high level. The village has a superior geographical location, a good overall human living environment, complete infrastructure, and a high level of economic development. There were 2 villages of this type, accounting for 2.08% of all left-behind villages: Jinhe village in Shuangwan town and Xipo village in Ningyuanbao town of Jinchuan District.

2. Improving short-board village: This type is mainly L-H-H-H, H-L-H-H, H-H-L-H and H-H-H-L type villages. The overall level of livability is high, and the resources and environment are good. Villages with significant shortcomings in the development process can improve the shortcomings in the development of projects and village space efficiency and achieve coordinated development of rural livability. There were 8 villages of this type, accounting for 8.33% of the total number of all left-behind villages. Among them, 75% of the improving short-board villages are in Jinchuan District.

3. Balanced developing villages: This type is mainly H-H-L-L, H-L-H-L, H-L-L-H, L-H-H-L, L-H-L-H, and L-L-H-H villages. The levels of livability in these types of villages vary. The government should continue to maintain the advantages, integrate and coordinate other functions, and develop villages in a balanced manner. There were 16 villages of this type, accounting for 16.67% of all left-behind villages, which were scattered throughout the study area.

4. Upgrading potential villages: This mainly includes H-L-L-L, L-H-L-L, L-L-H-L, and L-L-L-H type villages. The overall livability level is low. However, the individual livability shows advantages. The livable environment has a certain potential. The number of this type of village was the largest number, reaching 55 in total, accounting for 57.29% of all left-behind villages and accounting for 57.29% of all left-behind

villages. Among them, 96.36% of the upgrading potential villages are located in Yongchang County.

5. Comprehensive upgrading villages: This type is mainly L-L-L-L type villages. The livability levels for the elderly, women, and children, as well as the common livability levels, are low. The degree of rural subjects' aging and weakening is high. The hollowing is serious. The economic development is relatively poor. The basic service facilities and public services need to be further comprehensively improved. There were 15 villages of this type, accounting for 15.63% of all left-behind villages. They are located in Yongchang County.

## 6. Discussion and Conclusions

### 6.1. Discussion

6.1.1. Rural Livability under the Needs of the Left-Behind Population

The overall rural livability level in Jinchang was low, having the largest proportion of low-value areas. This indicates that the degree of satisfaction of the demands of the left-behind population was low. The overall economic development, public services, infrastructure, and configuration need to be optimized and improved. The living and production conditions need to be further improved because the society as a whole is oriented toward economic growth and is biased toward urban development, resulting in the emergence of rural left-behind populations [33]. Children, women, and the elderly, as the main groups of the left-behind populations, are facing changes in family structure, lifestyle, income, and attitude. Moreover, they have the shortcomings of insufficient labor capacity, high mental pressure, and low development vitality. This is a microcosm of the problems arising from China's urbanization and social transformation. As the vulnerable rural groups, they need more care and attention. The physiological needs of the left-behind population, like other rural residents, require good living conditions and infrastructure. Among them, the left-behind elderly often suffer from physical health problems due to factors such as high labor and living stress, unreasonable diet structure, and imperfect medical conditions [42,52]. Therefore, it is urgent to provide pension security and medical care. In terms of safety needs, left-behind women, as the key labor force of the family, have to raise children and support the elderly. They need a good employment environment to provide a stable income for the economic expenses of the family. In terms of social needs, the out-of-home work of some family members has greatly affected the economic foundation, life care, and spiritual world of the left-behind elderly. Left-behind women may lack a sense of security psychologically. Changes in the subjective and objective environment may cause their own psychological imbalances. Left-behind children face many problems such as growth risks, psychological burdens, and behavioral problems. They experience loneliness and other psychological pressures to varying degrees. All the "three-stay" populations need to belong in group interactions and public cultural and sports activities to satisfy their social needs. In terms of self-actualization needs, left-behind children may have problems such as fewer learning opportunities, low learning initiative, and a lack of control over moral conduct due to factors such as the lag in their grandparents' educational concepts and the lack of a family education environment and family emotions. School discipline and control can play an important role in guiding left-behind children in shaping their talents.

The orderly flow of urban and rural populations is an important way to reshape the urban pattern and reconstruct the rural space [53,54]. The movement of the rural population is for seeking a more livable environment and for better opportunities and living standards. Improvement of the level of rural livability can significantly affect the willingness of the floating population to stay [55]. This can reduce the outflow of talents [56] and take advantage of the process of introducing rural talents. This can guide the rational flow of the floating population and reduce the left-behind population. The degree of satisfaction of the development needs of the rural left-behind population is an important indicator of the rural livability level. Improvement of the rural livability level can improve the quality of life of

the left-behind population and provide a sense of gain, happiness, and safety in the process of rural development. The Strategic Plan for Rural Revitalization (2018–2022) emphasizes that it is necessary to promote public education, health care, social security, and other resources in rural areas, gradually establish and improve the basic public service system with universal coverage, universal sharing, and urban-rural integration, and promote the equalization of basic public services in urban and rural areas. Sharing the benefits of social development is caring for disadvantaged groups. This can promote equality of rights in rural areas and provide a harmonious social environment [57].

6.1.2. Discussion on the Classified Governance Path of Left-Behind Villages

1. Optimizing and upgrading village: This type of village has a good economy, complete infrastructure, and public services. These villages are located on the periphery of the urban area. Under the rural revitalization and development model of "leading rural areas with urban areas and promoting agriculture with industry", the government can promote the complementarity and flow of resources between urban and rural areas and give full play to the trickle-down effect of surrounding towns [21]. This can realize urban-rural integration and rural transformation and upgrading, speed up the process of interconnection and sharing of public services between urban and rural areas, and improve the quality of life of the left-behind population. Villages with rural tourism advantages can actively explore the integrated development of "tourism +" industries under the background of normalized epidemic prevention and control, protect rural characteristic resources, meet the demands of urban and rural residents' upgraded consumption, build a large-scale, professional, characteristic, and diversified rural tourism industry system, stimulate new vitality in the countryside, and create a village that is livable, recreational, and suitable for business and tourism.

2. Improving short-board village: The future development of this type villages focuses on making up for its shortcomings. Left-behind elderly villages can establish a diversified rural elderly care public service mechanism based on government financial subsidies. The government should gradually open up the elderly service market, coordinate social organizations, and innovate a method for elderly care. This can alleviate the problem of insufficient care for the left-behind elderly and guarantee the psychological and physical health of the left-behind elderly. In addition, the government should use the digital cultural stations in Jinchang to promote digital cultural experience services, enrich the spiritual lives of the left-behind population, retain cultural memories and cultural emotions, and improve the mental outlook of villagers. The government should pay attention to the value and potential of left-behind women in rural revitalization. Left-behind women villages can organize female deputies to people's congresses and cadres of women's federations to publicize in the villages. This can enhance the enthusiasm of left-behind women in social participation and protect the rights and interests of political participation. Giving full play to the role of women in rural governance can alleviate the conflicts in rural left-behind families and allow left-behind women to reshape their development value in the process of social participation. Left-behind children villages need to improve the education mechanism, with school as the core and the government, family, and village as the support. This can change the educational imbalance between urban and rural areas between regions and between schools.

3. Balanced developing villages: The livability evaluation of this type of village is at a medium level. The government can focus on improving the employment rate of rural industries based on village development and resources. The government can focus on developing modern industrial areas along national highways, creating advantageous brands, building information-based and specialized rural industries, enriching jobs, and realizing the employment transformation of left-behind women. Based on traditional agriculture, the government should focus on animal husbandry and plateau vegetables to develop characteristic industries and form an agricultural industry

system with planting, breeding, production, supply, and marketing [58]. The government should explore the "Internet + agriculture" model, extend the industrial chain of agricultural products, and promote the process of agricultural network sales.

4. Upgrading potential village: This type of village should ensure the allocation and supply of public service facilities, set up a comprehensive supporting system of rural basic public facilities, and optimize the spatial layout of the facilities based on the actual development needs of the left-behind population [59]. This can achieve the goal of equalizing urban and rural public services. Left-behind elderly and left-behind children villages can reorganize rural life circles according to the temporal and spatial behaviors and development needs of the left-behind population [60] and based on changes in village blood, geographical, and professional relationships. Improving the construction of schools and medical facilities can improve the completeness of facilities, provide educational and medical resources to the weak areas, and optimize the spatial layout. Schools can improve the construction of boarding systems to alleviate the problems of poor learning autonomy and the low safety and security of left-behind children. Left-behind women villages can develop rural industries in conjunction with towns and industrial areas to increase opportunities for left-behind women to be employed near home and train their labor skills. This can provide human resources for the development of rural industries, thereby improving the employment environment in rural areas and reducing the burden on left-behind families.

5. Comprehensive upgrading villages: This type of village is seriously aging and weakening, and the development vitality is insufficient. Under the background of population exodus, the construction of central villages within the town area can be strengthened in the future. The large-scale, professional, and market-oriented development of the rural economy can be promoted. The spatial structure and resource allocation can be optimized to drive the development of other surrounding villages. Comprehensive upgrading villages should further improve infrastructure, establish and improve basic public services and social security systems that coordinate urban and rural areas, benefit the whole village and urban and rural areas, and improve the quality of life of the left-behind population. This offers medical care for the left-behind elderly and education for left-behind children. The government should increase policy support and provide certain allowances to the left-behind population through the Spring Bud Program and Golden Talent Program.

6.1.3. Shortcomings and Outlook

To solve the problem of the rural left-behind population, the government should adjust the relationship between family members, help migrant workers return to their hometowns for employment, or move the left-behind population into cities and towns to live with migrant workers. However, there are other factors to consider, such as the fact that migrant workers do not have good economic conditions in urban areas and left-behind populations are unwilling to leave the countryside [42]. This hinders the solution of the left-behind problem. The urban-rural dual system, which is the reason for the occurrence of left-behind populations, will not disappear. However, the relationship between urban and rural areas is not an antagonistic relationship. The prosperity of the two should be integrated and developed on the basis of the equal status of urban and rural areas, the efficient allocation of factors, and the mutual promotion of the system [1]. The key factor of rural revitalization is people. The key to solving the problem of the left-behind population is to attract rural people to return to their hometowns for employment so that the structure of left-behind families is complete. To consolidate the achievements of poverty alleviation and improve the level of livability in rural areas, the government must solve the problem of brain drain in rural construction. By formulating relevant policies and incentive mechanisms for returning to their hometowns, the government can promote industrial development to retain talents and encourage more outstanding urban talents to go to the countryside to start businesses. The government should optimize the environment for innovation and



entrepreneurship in rural areas and increase the attractiveness of rural areas outside of blood, kinship, and geography. This can improve the quality of rural human capital, attract talents to return home, and retain local talents, thereby achieving the ultimate goal of urban-rural integration.

Based on the identification of the types of left-behind villages, this paper established a rural livability evaluation index system that combines the individual and common development needs of different subjects and analyzed the livability of left-behind villages in Jinchang city. Based on the combined characteristics of the left-behind population and livability, we constructed a classification basis to classify villages. The results of the study provide a reference for the construction of rural livability in left-behind villages. However, there are some weaknesses in this paper. First, this paper built an evaluation index system for the livability of an environment for the elderly, women, and children. However, due to the availability of village-level data, only the evaluation of the livability levels of villages and a study of the classified governance path were carried out. There needs to be further analysis about studying the perception and future demands of different left-behind groups for the livability of the left-behind environment and the construction goals of rural livability under the symbiotic states of different left-behind groups. Second, due to the lack of long-term survey data from 96 left-behind villages in Jinchang city, the conclusions drawn from the cross-sectional data analysis are limited, and the generalizability has to be verified by empirical studies in other similar areas.

*6.2. Conclusions*

This paper takes Jinchang city, Hexi Corridor, China as an example, identifying the left-behind types of villages by using rural survey data and evaluating rural livability by establishing the evaluation index system with "individuality + commonality". Combining the characteristics of the left-behind groups and livability, this paper derived different types of villages and proposed corresponding strategies for optimizing governance. The results show the following: (1) The rural left-behind villages in Jinchang city can be divided into four types: elderly dominant type, women-dominant type, children-dominant type, and comprehensive type. The left-behind children and left-behind elderly are the main types, accounting for 68.75% of the total number of left-behind villages. The left-behind villages were mainly concentrated in Yongchang County, accounting for 84.21% of all left-behind villages. (2) The livability values for the elderly, women, and children were low, mainly in the low-to-medium level. The average livability for children was the largest, and the average livability for women was the smallest. In addition, the livability levels for the elderly and children were mainly in the medium area, accounting for 48.95% and 46.88% of the total number of left-behind villages, respectively. The livability for women was mainly in the low-value areas, accounting for 60.42% of the total number of left-behind villages. (3) The overall livability level of the villages was low. The average of the livability index was 0.2336. The above-average evaluation units accounted for 40.625% of the total number of evaluation units. This indicates that the degree of meeting the needs of the left-behind population in villages is low. The overall economic development, public services, infrastructure level, and configuration need to be optimized and improved. The living and production conditions need to be further improved. (4) Based on the spatial differences and balance of individual livability levels and common livability levels in villages, we divided the villages into five types: optimizing and upgrading villages, improving short-board villages, balanced developing villages, upgrading potential villages, and comprehensive upgrading villages. In the future, it is necessary to carry out classified governance from multiple levels, thereby achieving the ultimate goal of sustainable rural development. Furthermore, the purpose of the carried-out research has a double meaning. On one hand, this research can bring theoretical and methodological contributions to rural governance, revitalization of left-behind villages, and sustainable oasis development. It can fill the gap in the research on the left-behind population in rural geography to a certain extent. On the other hand, the research results and the proposed governance path can promote

and solve the practical sustainable development problems of left-behind villages and help relevant departments to guide the care and prosperity of the left-behind population in practical work.

**Author Contributions:** L.M. and Y.Z. designed the study and processed the data. Z.S. and H.D. gave comments on the manuscript. All authors have read and agreed to the published version of the manuscript.

**Funding:** This paper is funded by two National Natural Science Foundation of China, whose grant number are 42101276 and 41961033, and Natural Science Foundation of Gansu Province (20JR5RA519).

**Institutional Review Board Statement:** Not applicable.

**Informed Consent Statement:** Not applicable.

**Data Availability Statement:** Not applicable.

**Conflicts of Interest:** The authors declare no conflict of interest.

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
