# Peer review of "The Rural Livability Evaluation and Its Governance Path Based on the Left-Behind Perspective: Evidence from the Oasis Area of the Hexi Corridor in China"

_sustainability, doi:10.3390/su14116714_

Round 1

Reviewer 1 Report

The article is correctly edited and based on correct assumptions and research methods. However, in order to improve it, I suggest:

- clearly define the purpose of the study - write it in the Introduction and Abstract
- in part 1. Introduction it’s needed to explain what extent the selected area as a case study is typical for this type of units in China, whether this area can be considered representative for the country.
- Part 3 Overview of the study area should be shorten - 3.1 Overview of physical geography should be expand 3.2 Socio-economic overview should be expand to include the context of the city's functional relationship with rural areas. In relation to the research topic, this aspect is more important than the physical geography features.
- in part 3.2 Socio-economic overview [ lines 221-225] provide data substantiating the information presented there or other sources of this information - research papers, reports.
- in part 6 Discusion and conclusion  author should explain whether the proposed scenarios have been applied and verified for other areas or whether they are the result of specific models or concepts of development confirmed scientifically
- 6.2. Conclusion -this part should be written differently or resign from this part as it is a repetition of content presented in part 5 Result analysis and part 6 Discussion and conclusion. I suggest presenting a response to the explicitly clearly specify purpose of the research (when will be assess).

Author Response

Dear Editors and Reviewers:

Thank you for your letter and for the reviewers' comments concerning our manuscription titled “The rural livability evaluation and its governance path based on the left-behind perspective:Evidence from oasis area of Hexi Corridor in China”(sustainability-1681497).Those comments are all valuable and very helpful for revising and improving our paper, as well as the important guiding significance to our researches. We have studied comments carefully and have made correction which we hope meet with approval. Revised portion are marked in the paper.The main corrections in the paper and the responses to the reviewer’s comments are in flowing documents.

Reviewer 2 Report

I found myself confused while reading your manuscript. While I certainly appreciate the need to address “the problems arising from China's urbanization and social transformation” (line 495), I was never sure how your approach would accomplish this goal.

First, you assume that the evaluation index you develop is helpful for achieving sustainable development, but you don’t ever demonstrate that it is. Rural livability is important but your focus within this broad construct isn’t clear. You define livability in several ways conceptually and then you operationalize the notion in opaque ways. For example, why would pension security vary among villages? Why are employment opportunities important for the livability for the elderly? (I understand why they lead to being left-behind but I don’t understand how they affect the quality of life once one is left behind.) In general, more explication of your approach would be helpful to the reader.

Second, you initially seem to promise the use of qualitative and quantitative methods (at line 74). But you never actually ask any of the people whose lives you are investigating anything at all, best I can tell. I do not know, therefore, how you can make any claims about the “degree of satisfaction of the demands of the left-behind population….” (line 487). The people you study come across as abstractions. Especially given your overly precise results (see section 5.2.1 of the manuscript) and your very general suggestions for improvement (see section 6.1.2), your manuscript seems to be more of a modeling exercise than an effort to really understand the needs of real people. Another way to put this is to ask what is the lived experience of livability; I don’t have a good sense of that after reading your manuscript.  

Third, I don’t really see the logic, or ultimately the utility, in your categorization of villages. Why use the 50% cutoff for those who have been left behind in terms of classifying a village? Why classify them at all? I assume that those villages with a plurality of elderly also have left behind women, no? Additionally, there is lack of clarity in the scheme because a woman of, say, 70 years old is both elderly and female.

Here are some specific comments (by line number):

38        The move from division to cooperation seems possible, but hardly universal nor assured.

48        Why is the key for achieving sustainability the construction of an evaluation index? Such an undertaking may be helpful, but you need to explain and show why and how.

85        You seem to imply that if it were possible to move these people to the city that would be preferred.

92        You should explain what you mean by “the period of social transformation.” Similarly, at line 124, in regard to the “household contract responsibility system”.

118      What is “individuality + commonality”?

156      Maslow seems extraneous. Why do you invoke him?

167      I don’t know whom you are referring to as “ordinary rural residents.”

175      Equal to whom?

248      I have many questions here: Why would this pattern exist? Why is it important? Why us 50% the right threshold. How could the proportions ever be exactly equal?

271      This statement about the dimensions listed in Table 1 needs much unpacking. You need to explain and justify these dimensions much better, and not simply assert them.

322      What does an entropy method mean?

340      I didn’t understand how “governance type” comes into the discussion nor what you mean.

432      This section seems very theoretical. Do you need actual empirical results to do this?

535      It seems to me that you might have known to make these recommendations even without your scheme and data.

Author Response

(The authors gave the same response as above.)

Reviewer 3 Report

Dear Authors,

The submitted manuscript titled „The rural livability evaluation and its governance path based on the left-behind perspective:Evidence from oasis area of Hexi Corridor in China” contains interesting findings, which might interest the international audience. The manuscript id generally well-written, however I have found some imperfections, which should be improved.

  1. I suggest to list the aims of investigations (or working hypotheses) at the end of chapter Introduction.
  2. Due to literature of subject is very large I encourage Authors to take into consideration the dividing the chapter Introduction into two separate subchapters: (i) The background and study aims and (ii) The present state of knowledge on livability of left-behind people in China
  3. Figures 4,5,7 are illegible., while Figure 6 in my opinion is not self-explanatory.
  4. Please, look into the following literature sources, perhaps some of them might be helpful in manuscript correction:
  • Li et al. 2015. The health of left-behind children in rural China. CHINA ECONOMIC REVIEW 36: 367-376. DOI10.1016/j.chieco.2015.04.004
  • Wang 2020. Left Behind by COVID-19 Experiences of "Left-Behind" Girls in Rural China. GIRLHOOD STUDIES-AN INTERDISCIPLINARY JOURNAL 13 (3) , pp.17-31.
  • Chang et al. 2019. Understanding the Situation of China's Left-Behind Children: A Mixed-Methods Analysis. DEVELOPING ECONOMIES 57 (1) , pp.3-35.
  • Hong and Fuller 2019. Alone and "left behind": a case study of "left-behind children" in rural China. COGENT EDUCATION 6 (1).
  • Liang et al. 2017. Depression among left-behind children in China. JOURNAL OF HEALTH PSYCHOLOGY 22 (14) , pp.1897-1905.
  • Wang et al. 2019. The personality traits of left-behind children in China: A systematic review and meta-analysis. PSYCHOLOGY HEALTH & MEDICINE 24 (3) , pp.253-268.

Author Response

Dear Editors and Reviewers:

Thank you for your letter and for the reviewers' comments concerning our manuscription titled “The rural livability evaluation and its governance path based on the left-behind perspective:Evidence from oasis area of Hexi Corridor in China”(sustainability-1681497).Those comments are all valuable and very helpful for revising and improving our paper, as well as the important guiding significance to our researches. We have studied comments carefully and have made correction which we hope meet with approval. Revised portion are marked in the paper. The main corrections in the paper and the responses to the reviewer’s comments are in flowing documents.

Reviewer 4 Report

Interesting study based on the well described theoretical framework (Figure 1. particularly), latest research articles and data are used. The authors have provided some visual information (maps, figures) thus readers can better understand the context and the results.

My suggestions to improve the paper are as follows:

1) It would be very useful to add some scientific and statistical data and evidence in section 1.2. on social-economic situation in rural China particularly emphasizing the rural situation in the studied area (as China is a huge country and most likely not all rural areas are similar). For the scientific paper, it is not enough to make a references only to political decisions such as CPC National Congress in terms of justification the problem statement; necessity for the research must be based primarily on the academic evidence.

2) In the section 4.1. a survey is mentioned as a data source; however, it is not clear, what kind of survey it was and what kind of data were collected, what was the fieldwork of data collection, sample description is needed. Some statements about the research ethics is also needed. This information must be added in the research methodology part.

3) Conclusions are derived from the research. The first conclusion actually is not a conclusion. This information better fits in introduction; here the authors can reflect whether they have reached their aim and how. Conclusion part must include some discussion with the literature reviewed in the theoretical part of the study.

Author Response

(The authors gave the same response as above.)
